# Exploring Immune-Related Gene Profiling and Infiltration of Immune Cells in Cervical Squamous Cell Carcinoma and Endocervical Adenocarcinoma

**DOI:** 10.3390/genes15010121

**Published:** 2024-01-19

**Authors:** Jialu Li, Juqun Xi

**Affiliations:** 1School of Medicine, Institute of Translational Medicine, Yangzhou University, Yangzhou 225012, China; 212203112@stu.yzu.edu.cn; 2Jiangsu Key Laboratory of Integrated Traditional Chinese and Western Medicine for Prevention and Treatment of Senile Diseases, Yangzhou 225009, China

**Keywords:** immune related genes, *CXCL*8, *CXCL*10, immune infiltration, CESC

## Abstract

Cervical cancer is a widespread malignancy among women, leading to a substantial global health impact. Despite extensive research, our understanding of the basic molecules and pathogenic processes of cervical squamous cell carcinoma is still insufficient. This investigation aims to uncover immune-related genes linked to CESC and delineate their functions. Leveraging data from the GEO and ImmPort databases, a total of 22 immune-related genes were identified. Multiple tools, including DAVID, the human protein atlas, STRING, GeneMANIA, and TCGA, were employed to delve into the expression and roles of these immune genes in CESC, alongside their connections to the disease’s pathological features. Through RT-PCR, the study confirmed notable disparities in *CXCL*8 and *CXCL*10 mRNA expression between CESC and normal cervical tissue. The TCGA dataset’s immune-related information reinforced the association of *CXCL*8 and *CXCL*10 with immune infiltration in CESC. This research sheds light on the potential of *CXCL*8 and *CXCL*10 as promising therapeutic targets and essential prognostic factors for individuals diagnosed with CESC.

## 1. Introduction

Cervical cancer is a prevalent global issue, ranking as one of the most widespread cancers within the human body [1]. Persistent HPV infection induces immune tolerance in the host immune system, which is one of the important mechanisms of cervical lesions. Other important factors include sexually transmitted infections, smoking, giving birth to a large number of children, and long-term use of oral contraceptives [2]. The burden of treating cervical cancer in developing nations remains high due to the absence of effective preventive measures, like HPV vaccination and screening [3]. In recent years, epigenome and chromatin remodeling modifiers have become the preferred molecules for regulating cancer cell responsiveness to specific therapies. Many targeted molecules are being attempted to be developed as anticancer agents for the treatment of cervical cancer [4]. Despite current advancements in medical treatment and diagnosis, the outlook for patients is still far from promising. Cervical cancer encompasses two main types: cervical squamous cell carcinoma (CESC) (75% of cases) and cervical adenocarcinoma (25% of cases) [5]. Therefore, deepening the understanding of the pathogenesis of CESC, proposing new therapeutic targets and methods, is crucial for early diagnosis of the disease. Currently, the assessment and prediction of Cesc management and prognosis heavily rely on the updated 2018 staging guidelines provided by the International Federation of Gynecology and Obstetrics (FIGO) [6]. Numerous studies have investigated the connection between cervical cancer and immune genes, showing promising results in CESC treatment by targeting CTLA4 and PD1 agents in immunotherapy [7,8]. In order to further improve the overall survival rate of CESC patients, it is crucial to have a deeper understanding of the root causes of CESC.

The newly discovered observations reveal the important role of chemokines (*CXCL*) in the tumor microenvironment of cervical cancer. *CXCL*1-17, a small protein released by tumor cells, immune cells, and stromal cells, plays a pivotal role [9]. Interestingly, *CXCL*15 expression has been observed in mouse organs but lacks a human equivalent. Based on the presence of the glutamic acid leucine arginine (ELR) motif, *CXCL*s can be mainly classified into two groups: ELR+ *CXCL*s, which engage with neutrophils, fostering angiogenesis; and ELR- *CXCL*s, primarily binding to lymphocytes, inhibiting angiogenesis [10]. These *CXCL*s function by binding to specific CXC chemokine receptors (CXCRs), generally G-protein-coupled receptors spanning seven membranes, which activate various signaling cascades [11]. CXC chemokines exert promotive or suppressive effects on tumors by recruiting different immune cell types to specific tumor sites. Furthermore, chemokines can influence tumor cell proliferation, metastasis, and stromal cell biology, participating in the generation and metastasis of tumor vessels [12,13,14].

Prior investigations have underscored the therapeutic and prognostic significance of *CXCL* in diverse cancer types, including pancreatic, colon, prostate, and glioblastoma [11,15,16]. Studies have indicated that elevated *CXCL* levels in glioblastoma hinder the DNA damage response and foster tumor progression [17]. *CXCL* has been associated with survival outcomes in patients with colorectal cancer. Similarly, the increased expression of *CXCL*1/2/8 proteins in cervical cancer can drive tumor promotion by stimulating cancer cell proliferation and angiogenesis through an autocrine mechanism involving CXCR2 in endothelial cells [18]. Conversely, inhibiting *CXCL*8 curtails cervical cancer cell proliferation and induces cancer cell apoptosis [19]. Hence, *CXCL*s may play a pivotal role in the onset and progression of CESC. The aim of this study was to evaluate the expression and function of immune genes in CESC, using bioinformatics and molecular biology.

## 2. Methods

### 2.1. Data Collection

We collected four microarray datasets (GSE122697, GSE9750, GSE64217, and GSE7410) from the GEO database, using the keywords “cervical cancer”. Specifically, GSE122697 comprised 11 normal cervical specimens and 5 CESC specimens, while GSE9750 had 33 adjacent normal cervical specimens and 24 CESC samples. GSE64217 included 4 normal cervical samples and 2 CESC samples, and GSE7410 contained 40 normal cervical samples and 5 CESC samples. We filtered the data with the criteria: condition, |log2| ≥ 1.0, and *p* < 0.05. Additionally, we downloaded CESC patient samples and relevant clinical information from the TCGA database.

To identify differentially expressed genes (DEGs) between the CESC and control groups, we used the “limma” package in R software (Version R-4.2.2), setting a significance threshold of *p*-value < 0.05. The DEGs were visualized using volcano plots.

For acquiring immune-related genes, we utilized the immune gene set cross-overlap from the ImmPort database (https://immport.niaid.nih.gov, accessed on 1 January 2020), aligning them with the DEGs obtained from the GEO database.

Gene ontology (GO) analysis and pathway enrichment analysis are often performed on large-scale transcriptional or genomic data. We utilized the pathway databases from the KEGG (Kyoto Encyclopedia of Genes and Genomes), which is an invaluable resource that provides information about genes and molecular networks, for genetic studies. The filtered DEGs underwent GO and KEGG analysis, and we utilized the David tool to retrieve GO and KEGG annotations for DEG participation. A significance threshold of *p*-value < 0.05 was applied for the screening of GO and KEGG analysis. R (https://www.r-project.org/, accessed on 1 January 2020) served as the primary tool for conducting the analysis.

### 2.2. Pathological Analysis

Input the immunohistochemical profiles of proteins in cervical tissues of normal women of the same age and cervical lesions of cancer patients into the Human Protein Atlas (HPA) database. The sample is stored in freezing mode.

We created a network of protein–protein interactions (PPIs) involving the genes that exhibited differential expression using the interaction gene retrieval database (STRING) to understand the interactions among the proteins encoded by these genes. To visualize this biological network and the integrated string database, we utilized Cytoscape (version 3.7.2). We considered a combined score above 0.5 as statistically significant and identified hub genes within this PPI network, using the CytoHubba application.

### 2.3. Cell Culture

Human CESC cell lines (HeLa, hela299, and ht-3) and a normal cell line (H8) were cultured in RPMI-1640 medium supplemented with 10% fetal bovine serum under incubation conditions at 37 °C with 5% CO_2_. Total RNA was extracted from the cardiomyocytes, using the rnaiso plus kit, and the concentration and quality of the extracted RNA were assessed spectrophotometrically. The presence of 28S and 18S bands was observed on a 1% agarose gel stained with ethidium bromide. cDNA synthesis was conducted using the Script RT reagent kit, followed by incubation with extaqmII at 20 °C to measure the mRNA expression levels of *CXCL*8, *CXCL*10, and GAPDH. A 25 μL PCR mix was prepared, comprising 12.5 μL SYBR, 1 μL forward primer (10 μM), 1 μL reverse primer (10 μM), 1 μL cDNA, and 9.5 μL ddH2O. The PCR reaction program included an initial denaturation step at 95 °C for 5 min, followed by 40 cycles of denaturation at 95 °C for 10 s, annealing at 60 °C for 30 s, and extension at 72 °C for 5 min. Amplification efficiency of each gene was validated by generating a standard curve based on four steps of cDNA dilution. All samples were analyzed in triplicate, and quantification was performed by mean cycle threshold, using the 2^△△^CT method previously described. The primer sequence is shown in Table 1.

### 2.4. Immune Infiltrate Analysis

We collected DNA from the UCSC (https://xenabrowser.net/, accessed on 1 January 2020) and acquired a uniformly normalized dataset of cervical cancer from the TCGA database. From the respective samples, we isolated the expression data of the genes *CXCL*8 (ENSG00000169429) and *CXCL*10 (ENSG00000169245). We filtered the sample sources to include only TCGA-CESC samples. We applied a log2 (x + 1) transformation to each expression value. Then, we mapped the gene expression profiles of CESC onto the gene symbols. Using R software (version 1.0.13), we calculated stromal, immune, and estimated fractions for each patient in each tumor based on gene expression.

For statistical analysis, we used IBM SPSS 21.0 and graphpadprism9 software. Normally distributed measurements are presented as mean ± SD, while outlier measurements are shown as median (25th–75th percentiles). Significance levels are denoted as * *p* < 0.05, ** *p* < 0.01, and *** *p* < 0.001.

## 3. Results

### 3.1. Differential Gene Screening

We analyzed immune-related genes that exhibited a differential expression in the CESC bioinformatics datasets. We sourced four microarray datasets (GSE122697, GSE9750, GSE64217, and GSE7410) from the GEO database. Among these, the GSE122697 dataset showed 1006 DEGs (2242 upregulated and 1908 downregulated). In the GSE9750 dataset, we identified a total of 741 upregulated and 2083 downregulated genes. Additionally, in the GSE64217 dataset, we found 3825 upregulated and 3560 downregulated genes. Moreover, in the GSE7410 dataset, we found 935 upregulated and 3408 downregulated genes. The resulting DEGs of all four datasets are depicted in Figure 1.

### 3.2. Immune Gene Acquisition

The amalgamation of the four datasets through Venn diagram analysis led to the identification of a collective 186 DEGs (Differentially Expressed Genes). Within this set, 122 genes displayed upregulation, while 64 genes exhibited a downregulation (as illustrated in Figure 2A,B). The ImmPort database contained a total of 2483 immune-related genes. Interestingly, a subset of 22 immune genes was found within the 186 DEGs, identified through the intersection of Venn diagrams (Figure 2C,D). Among these, 14 genes were upregulated, and 8 genes were downregulated.

### 3.3. Pathological Analysis of HE Staining in Cervical Carcinoma

Figure 3A,B display data obtained from the HPA library, showcasing a comparison between the cervix of healthy females and those with cervical cancer. In contrast to the normal cervical tissue, cancerous cells exhibited notable differences, including an increased cell count, significantly enlarged nuclei, hyperchromatic nuclei, a noticeable enhancement in chromatin staining intensity and thickness, and a distinctive bluish-purple droplet-like feature. These observations indicate a highly malignant and proliferative state of the cancer cells.

### 3.4. Immunohistochemical Results of Immune Genes in Cervical Cancer

The immunome data for immune-related genes in both normal tissues and cervical cancer tissues were obtained from the HPA. At the junction of squamous epithelium and columnar epithelium at the outer cervical opening, samples are taken for biopsy and analysis in areas with deep or special lesions to the naked eye, as depicted in Figure 4A-1–R-2. Specifically, the expression levels of *CXCL*8, IDO1, STAT1, ISG15, PLSCR1, RSAD2, BIRC5, EDN2, TYMP, IL32, SPP1, *CXCL*12, PTGDS, SPINK5, PIK3R1, SLIT2, ESR1, and TGFBR3 were analyzed in these tissue types. In the context of this analysis, a value of 1 represents normal tissue, while a value of 2 signifies cancerous tissue. Notably, *CXCL*8, IDO1, STAT1, ISG15, PLSCR1, RSAD2, BIRC5, EDN2, TYMP, IL32, and SPP1 exhibited significantly higher expression in cancer tissues, whereas *CXCL*12, PTGDS, SPINK5, PIK3R1, SLIT2, ESR1, and TGFBR3 showed a moderate level of expression in cancer tissues.

### 3.5. Functional Enrichment Analysis

Cluster analysis tools were applied to investigate 22 immune-related genes with a differential expression (DEGs). Both GO and KEGG analyses were utilized. The GO analysis revealed an extensive range of biological processes (BPs), cellular components (CCs), and molecular functions (MFs). Specifically, there were 700 BP, 27 CC, and 87 MF associated with these DEGs. Noteworthy enrichments were observed in key processes, such as the “Pathway of signaling mediated by cytokines”, “Activity of ligands binding to receptors”, and the “Outer surface of the plasma membrane”, as highlighted in Figure 4A-1–C-2, representing the top seven BPs, CCs, and MFs identified in this analysis.

Figure 5D, representing the KEGG analysis results, elucidates the key signaling pathways pivotal in the immune processes of cervical cancer. The analysis indicates that the “toll-like receptor signaling pathway” and the “chemokine signaling pathway” emerge as the primary routes contributing to the immune response in this context. This comprehensive exploration of immune-related DEGs and their associated pathways enhances our understanding of the immune mechanisms at play in female cervical cancer.

### 3.6. PPI Analysis of DEGs

In order to delve deeper into the existing interactions among these 22 immune-related DEGs, a comparative analysis was performed between the CESC and N groups. Subsequently, we constructed a protein–protein interaction network, encompassing 22 nodes and 160 edges (depicted in Figure 6A). Using the CytoHubba plug-in, hub genes significantly linked to CESC were identified based on a connectivity score threshold of 5 in the PPI network analysis. The top 10 gene clusters were identified using the degree, MCC, and MNC modes, all of which consistently included the following 10 genes: RSAD2, ISG15, *CXCL*9, IDO1, *CXCL*12, SPP1, *CXCL*8, ESR1, *CXCL*10, and STAT1 (illustrated in Figure 6B–D). Notably, *CXCL*8 and *CXCL*10 emerged as the most densely interacting genes across all three patterns (Figure 6B–D). Red, orange, and yellow nodes represent denser interactions that gradually weaken (Figure 6B–D).

### 3.7. CXCL8 and CXCL10 Interaction Network Construction and Gene Ontology

We employed the “gene MANIA” tool to construct a set of 22 genes related to *CXCL*8 and *CXCL*10. The protein–protein interaction (PPI) and co-expression network exhibited a total of 1423 connections, as depicted in Figure 7A. The gene ontology analysis of this network unveiled that these 22 genes were predominantly enriched in seven distinct biological processes, seven cellular components, and seven molecular functions, as illustrated in Figure 7B–D. Notably, *CXCL*8 and *CXCL*10, playing significant roles in immune function, may participate in the “chemokine mediated signaling pathway” and exhibit “chemokine activity”. Moreover, they could be associated with the “external side of plasma membrane”, suggesting substantial alterations in the outer plasma membrane, potentially contributing to the pathogenesis of CESC patients.

### 3.8. Expression Analysis of CXCL8 and CXCL10 in CESC

We employed RT-PCR to ascertain the mRNA expression levels of *CXCL*8 and *CXCL*10 in human cell lines derived from cervical cancer. In comparison to H8 (representing normal cervical cells), we observed a significant increase in *CXCL*8 and *CXCL*10 levels within the HeLa, hela299, and ht-3 cell lines (all being cervical cancer cells) (*p* < 0.01) (as shown in Figure 8A,B). Furthermore, this phenomenon was validated by analyzing the expression of *CXCL*8 and *CXCL*10 in both normal and cervical cancer tissues (as depicted in Figure 8C,D).

### 3.9. The Role of CXCL8 and CXCL10 in Immune Infiltration within CESC

Using the R package deal psych (model 2.1.6), we employed the test function to compute the Pearson’s correlation coefficient between gene expressions and immune infiltration scores within individual tumors. This allowed us to identify significant associations with immune infiltration, specifically observing a strong correlation in CESC for *CXCL*8 and *CXCL*10 (see Figure 9). The stromal scores exhibited the range (−0.2, 0.41), with a significant *p*-value of <0.001, and the immune scores range was (−0.22, 0.70), also with a significant *p*-value of <0.001. Furthermore, the estimate scores were within the range of (−0.24, 0.65), with a *p*-value of <0.001. The red dot represents individual data, the black line represents the trend between two variables on the x-axis and y-axis, and the yellow histogram represents the distribution of data on the x-axis.

## 4. Discussion

Cervical cancer stands as the second most prevalent malignancy among women, placing a considerable burden on both health and economic aspects on a global scale [20]. Among cervical cancer cases, CESC represents the primary pathological subtype. Alarming statistics reveal that, in certain regions, the 5-year survival rate for advanced cervical cancer is below 50% [21]. This has prompted oncologists to actively seek out therapeutic targets and predictive biomarkers, aiming to enhance the survival rates of CESC patients. These efforts involve the investigation of novel genes, mRNA, and proteins [22].

Recent years have seen a surge in interest in tumor immunity, with multiple studies highlighting the close connection between tumor development and the immune microenvironment [23]. In the context of this research, we embarked on a comprehensive bioinformatics approach to explore the potential significance of immune genes as both therapeutic targets and prognostic biomarkers for individuals afflicted by CESC.

Therefore, we employed bioinformatics techniques to integrate a gene set based on the expression profiles of immune-related genes. This set comprised 14 genes showing increased expression and 8 genes with decreased expression. We then analyzed the differences in pathological features between normal and CESC samples, using HE staining. Additionally, we examined the protein expression of several genes (*CXCL*8, IDO1, STAT1, ISG15, PLSCR1, RSAD2, BIRC5, EDN2, TYMP, IL32, SPP1, *CXCL*12, PTGDS, SPINK5, PIK3R1, SLIT2, ESR1, and TGFBR3), revealing notable differences between normal individuals and CESC patients. We selected the overlapping genes for further investigation.

Our GO and KEGG analyses demonstrated significant enrichment in the “cytokine-mediated signaling pathway”, “receptor-like activity”, and “external side of the plasma membrane”. Furthermore, the KEGG analysis indicated that the main signaling pathways involved in the immune process of female cervical cancer included the toll-like receptor signaling pathway, chemokine signaling pathway, and other signaling pathways.

Research has demonstrated that abnormal *CXCL* expression significantly impacts the development and advancement of tumors [24]. The presence of *CXCL*8 protein in the serum of colorectal cancer patients is notably higher than that in healthy control histones, influencing tumor stage and metastasis in these patients [25]. *CXCL*s are known to exert their effects on tumors through both direct and indirect means. For instance, *CXCL*9/10/11 can autonomously bind to their CXCR3a protein, thereby directly promoting the growth and metastatic properties of cancer cells [26]. Additionally, cervical cancer cells with elevated *CXCL*3 expression can enhance the autonomous proliferation and migration abilities of cancer cells, leading to an overall increase in cell count [27]. On the other hand, *CXCL*s primarily facilitate the chemotaxis of immune cells. *CXCL*9/10/11 can recruit immune cells, including CD8+ T cells, Th1 cells, and NK cells, to reach tumor sites, thereby inhibiting the progression of colorectal cancer through the binding of CXCR3 protein to immune cell surface sites [28].

In our study, a significant correlation was observed between elevated levels of *CXCL*8 and *CXCL*10 and reduced overall survival rates among individuals diagnosed with CESC, suggesting that these molecules hold promise as important prognostic indicators for this disease. Previous research has also highlighted the prognostic significance of *CXCL*s in different types of human cancers. Notably, in advanced high-grade serous ovarian cancer, patients presenting with an increased expression of *CXCL*9/10 demonstrated improved clinical outcomes [28]. Likewise, increased levels of *CXCL*10, *CXCL*12, and *CXCL*14 in hepatocellular carcinoma patients correlated with favorable survival rates [29]. However, our investigation revealed that the heightened expression of *CXCL*8 and *CXCL*10 led to reduced overall survival in CESC patients, aligning to our initial expectations.

Interestingly, we observed a noteworthy correlation between the expression levels of the *CXCL*8 and *CXCL*10 genes and the stromal score, immune score, and evaluation score in patients with CESC (see Figure 9). The tumor microenvironment is paramount for both tumor formation and progression [30]. The estimation algorithm utilizes an enrichment analysis on the gene set of the target sample, yielding three distinct scores: the matrix score, immune score, and estimation score. These scores aid in the analysis of the tumor tissue matrix, the assessment of immune cell infiltration, and the inference of tumor purity, respectively [31].

Investigating potential therapeutic targets related to immunity holds significant promise for reshaping the tumor microenvironment and restraining tumor metastasis. Numerous studies have highlighted the active role played by the immune microenvironment in the treatment and prevention of cancer [32]. Our comprehensive analysis of CESC data acquired from the TCGA database provides further evidence supporting the advantageous prognostic significance of immune-related genes that are present within the tumor microenvironment (TME) specifically for patients with cervical cancer. These findings highlight the potential of leveraging immune-related genes as valuable biomarkers for prognostic assessment and as potential therapeutic targets in the management of CESC.

Furthermore, the survival outcomes of patients with cervical squamous cell carcinoma (CESC) are intricately connected to factors within the tumor microenvironment (TME), such as the stroma, immune infiltration, and tumor purity. Previous research has underscored the significance of the immune microenvironment in the initiation and progression of tumors [33]. Furthermore, our comprehensive analysis of CESC patients’ data obtained from the TCGA database uncovers the substantial impact of immune components present within the tumor microenvironment (TME) on the overall prognosis. It is important to highlight that not only the stromal components, but also the immune infiltrates and tumor purity within the TME, hold noteworthy relevance to the development and progression of CESC, emphasizing the need for further extensive exploration in this area.

## 5. Conclusions

Thus, the study findings underscore the significance of leveraging bioinformatics approaches to gain insights into the aberrant functioning of immune genes in individuals with CESC, which could ultimately provide useful avenues for exploring potential therapeutic interventions and preventive strategies for the benefit of CESC patients. It identified that elevated levels of *CXCL*8 and *CXCL*10 may serve as potential markers for predicting reduced survival among CESC patients. Furthermore, further fundamental experiments are essential to delve into the precise functions and underlying molecular mechanisms of *CXCL*8 and *CXCL*10 in the context of CESC. Our research findings provide important assistance to cervical cancer patients, further enhancing our understanding of cervical cancer and contributing to clinical prognosis and diagnostic applications.

## Figures and Tables

**Figure 1 genes-15-00121-f001:**
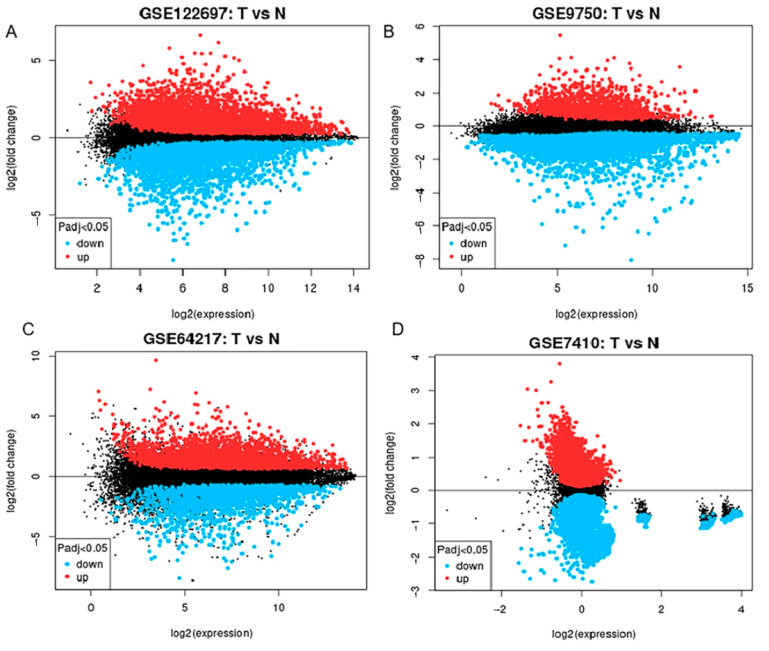
Displays a volcano map representing the csc anomaly dataset. This map contains four data sets: (**A**) GSE122697, (**B**) GSE9750, (**C**) GSE64217, and (**D**) GSE7410. In this map, the red color indicates genes that are upregulated as determined by log fold change (log FC) >1.0 and adjusted *p*-value < 0.05. Genes shown in black are downregulated and meet the criteria of log FC less than −1.0 and adjusted *p*-value less than 0.05. “T” represents samples from cervical cancer, and “N” represents samples from normal subjects.

**Figure 2 genes-15-00121-f002:**
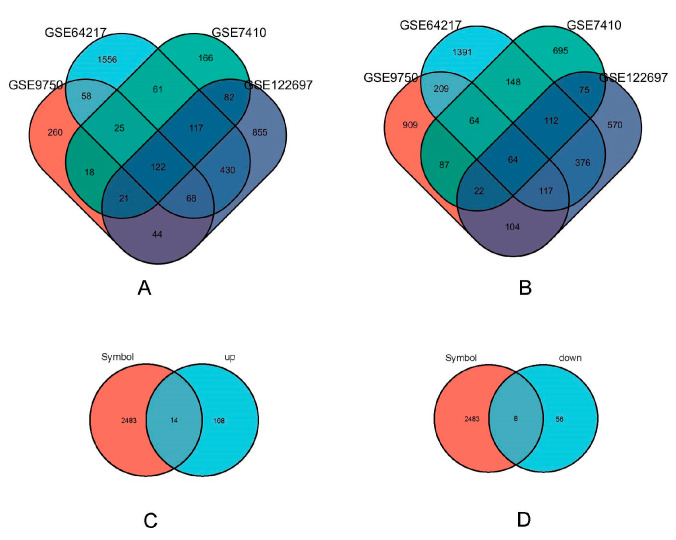
Immune related gene acquisition. (**A**) GSE122697 (purple), GSE9750 (orange), GSE64217 (blue), and GSE7410 (green) upregulated differentially expressed genes’ Venn diagram. (**B**) GSE122697 (purple), GSE9750(orange), gse64217 (blue) and GSE7410 (green) downregulated differentially expressed genes’ Venn graph. (**C**) Venn plot of immune genes (orange) in ImmPort database against the upregulated differentially expressed genes (blue) screened; (**D**) Venn plot of immune genes (orange) in ImmPort database against the downregulated differentially expressed genes (blue) screened.

**Figure 3 genes-15-00121-f003:**
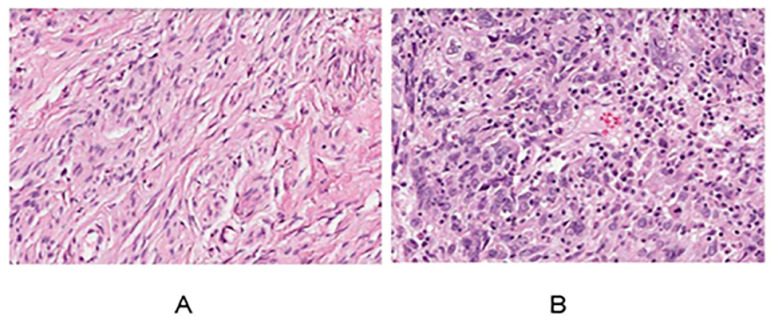
Normal cervix (**A**) and cervical cancer HE results (**B**) for women.

**Figure 4 genes-15-00121-f004:**
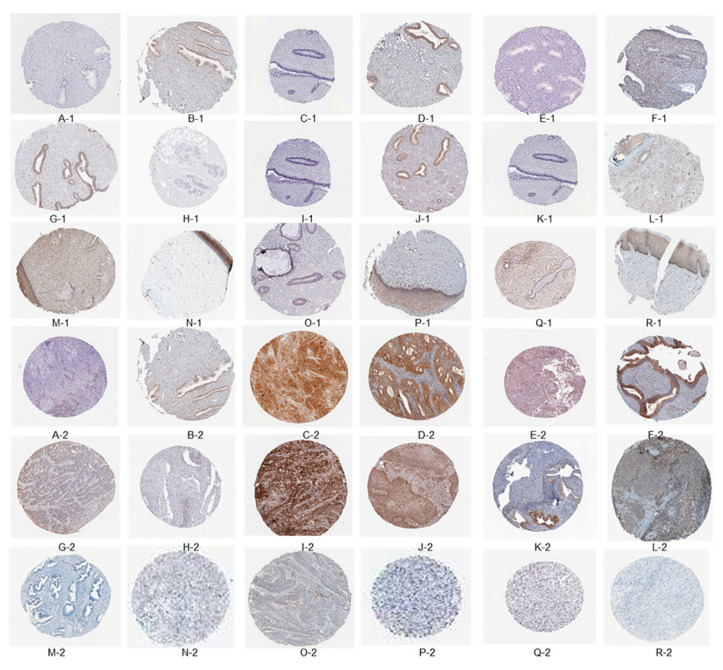
The immuno-histochemical findings indicate the presence of specific biomarkers in the examined tissues. The following biomarkers were detected: *CXCL*8, IDO1, STAT1, ISG15, PLSCR1, RSAD2, BIRC5, EDN2, TYMP, IL32, SPP1, *CXCL*12, PTGDS, SPINK5, PIK3R1, SLIT2, ESR1, and TGFBR3. The tissues were categorized as follows: tissue type 1 corresponds to normal tissue, while tissue type 2 signifies cancerous tissue.

**Figure 5 genes-15-00121-f005:**
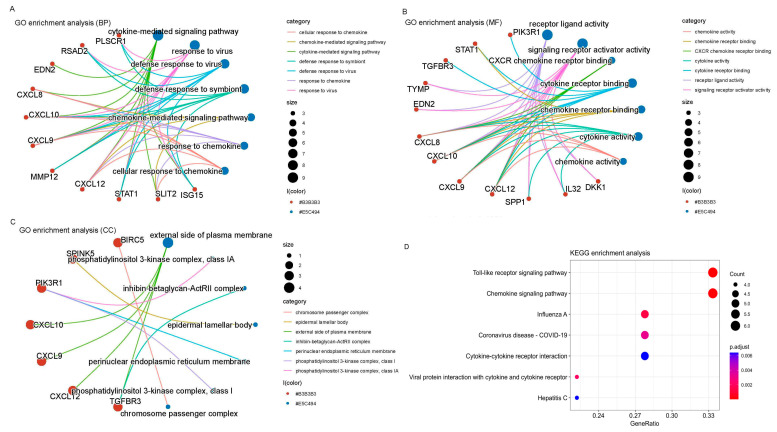
Gene ontology and KEGG enrichment analysis. (**A**) BPs. (**B**) CCs. (**C**) MFs. (**D**) KEGG enrichment analysis.

**Figure 6 genes-15-00121-f006:**
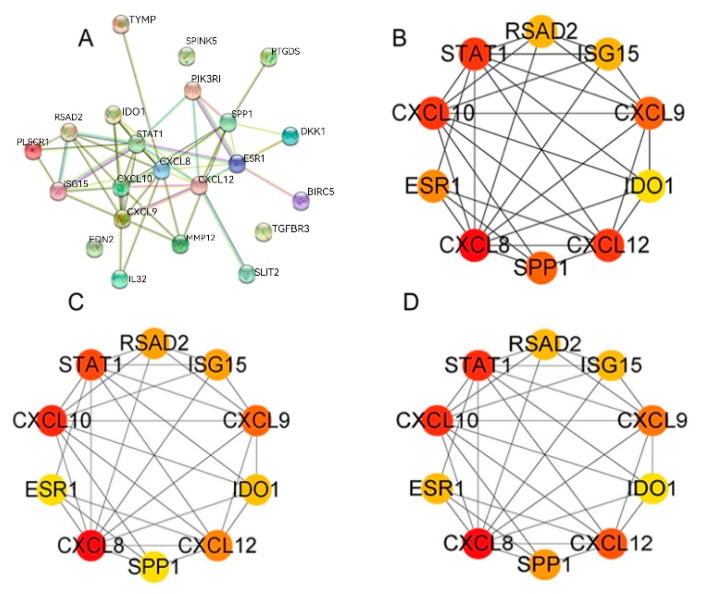
PPI network analysis of immune genes. (**A**) PPI network. (**B**–**D**) For the top 10 most significant gene clusters in the degree, MCC, and MNC modes.

**Figure 7 genes-15-00121-f007:**
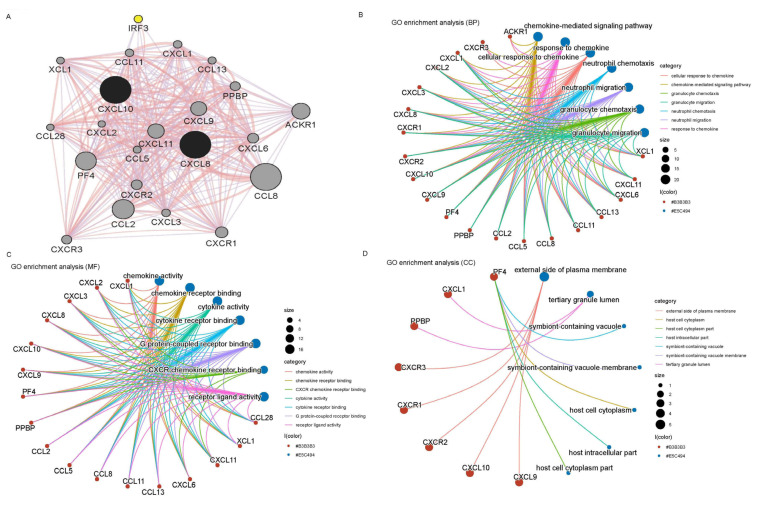
Putative *CXCL*8 and *CXCL*10 PPI networks and GO analysis. (**A**) Co-expression and PPI network. (**B**) Bioprocess. (**C**) Cellular components. (**D**) Molecular function.

**Figure 8 genes-15-00121-f008:**
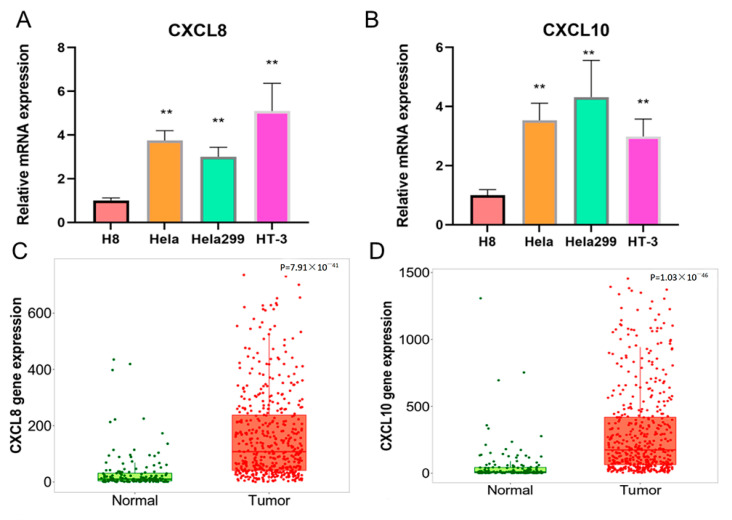
Different expression levels and bioinformatics analysis of *CXCL*8 and *CXCL*10 in cervical cancer cell lines. (**A**,**B**) The mRNA expression levels of *CXCL*8 and *CXCL*10 in hela, hela299, HT-3, and H8. (**C**,**D**) The gene expression of *CXCL*8 and *CXCL*10 in CESC tissues.The symbol “**” indicates significant differences.

**Figure 9 genes-15-00121-f009:**
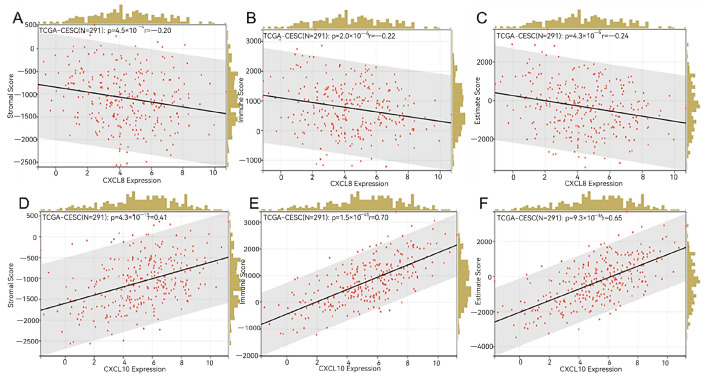
Immune infiltration analysis. (**A**–**C**) Stromal score, immune score, and estimate score of *CXCL*8 in CESC. (**D**–**F**) Stromal score, immune score, and estimate score of *CXCL*10 in CESC.

**Table 1 genes-15-00121-t001:** Primer sequences.

Gene		Primer Sequences
GAPDH	F	GGAGCGAGATCCCTCCAAAAT
	R	GGCTGTTGTCATACTTCTCATGG
*CXCL*8	F	TTTTGCCAAGGAGTGCTAAAGA
	R	AACCCTCTGCACCCAGTTTTC
*CXCL*10	F	GTGGCATTCAAGGAGTACCTC
	R	TGATGGCCTTCGATTCTGGATT

## Data Availability

The datasets generated/analysed during the current study are available.

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
