# Peer review of "Exploring Immune-Related Gene Profiling and Infiltration of Immune Cells in Cervical Squamous Cell Carcinoma and Endocervical Adenocarcinoma"

_genes, 2024, doi:10.3390/genes15010121_

Round 1

Reviewer 1 Report

Comments and Suggestions for Authors

Dear authors , thank you for your paper

The idea is very interesting and the paper very well written, but, unfortunately, I cannot accept your paper for publication.

Please, find my comments bellow:

Line 11: doesn’t make sense, please, re-write

Line 32:  Please, put the CESC in brackets the first time you use it. And always capitals

Lines 37-38: this statement is not accurate; surgery and/or radiotherapy can significantly improve survival for patients with cervical CA (especially in early stages)

Line 38: again, capitals

Lines 2-4: in the title you mention cervical SCC and adenoCA but all the paper is focused only on the SCC. Please, amend the title

Line 41: cxcl15, capitals, please

Line 62: your study aims to assess the expression of specific genes in patients with cervical cancer and to find a connection with patients’ survival. To my opinion, the first target is fully achieved but not the second one. In your methodology, there is not a single word regarding how you are going to assess patients’ survival. In the results session, line 241, you make a comment that patients with high expression of CXCL8 and CXCL10 have significantly shorter overall survival. There is no evidence behind this statement in your paper. When we are talking about survival it is vital to state: number of patients/ FIGO stage / treatment they received between the patient in the high vs low  CXCL8 and CXCL10 groups. If you cannot provide all these information to support the difference in survival, please amend the aim of your study.

Line 287: similar to the previous comments “ we found that CESC patients with high expression of CXCL8 and CXCL10 were significantly associated with worse overall survival”. How did you find it, show me the evidence about it!

thank you, looking forward to read the 2nd version

Reviewer 2 Report

Comments and Suggestions for Authors

It has exciting content, but there are many typographical errors throughout. It is also difficult to understand the context, and why the focus is on CXCL is unclear. Also, the text in Figures 5 and 7 is faded and unreadable; it is unclear where the immunostaining in Figure 4 is stained and what part of the tumor was biopsied.

The first sentence of the abstract is unclear. Is this because it was copied and pasted? Abbreviations are suddenly appearing.

In the introduction, the capitalization of words and phrases is uneven; CXCL is mentioned abruptly. The connection from the general cervical cancer story is unclear.

What tissue and from what site was method 2.2 taken from a properly consenting patient? What is the method of preservation? The age and stage of the disease are also unknown. The results of genetic analysis will vary greatly depending on this background.

Cell experiments, but Hela and Hela299 have the exact origin.

Overall, the paper is immature. We expect the paper to be resubmitted.

Comments on the Quality of English Language

There are many typographical errors throughout. It is also difficult to understand the context. Please make sure you know what you want to investigate through the research.

Author Response

请参阅附件。

Round 2

Reviewer 1 Report

Comments and Suggestions for Authors

Dear authors 

thank you for the revised version and excellent work you have done to improve the paper

minor changes:

Line 30: cervical with small “c”, not capital

Line 31: CESC capitals, please

Line 202: please, delete “female”

Congratulations for your paper

Reviewer 2 Report

Comments and Suggestions for Authors

I have no concerns related to this manuscript.